# High Speed Railway Fastener Defect Detection by Using Improved YoLoX-Nano Model

**DOI:** 10.3390/s22218399

**Published:** 2022-11-01

**Authors:** Jun Hu, Peng Qiao, Haohao Lv, Liang Yang, Aiguo Ouyang, Yong He, Yande Liu

**Affiliations:** 1School of Mechatronics & Vehicle Engineering, East China Jiaotong University, Nanchang 330013, China; 2School of Mechanical Engineering, Zhejiang University, Hangzhou 310027, China

**Keywords:** track fastener, feature fusion, attention mechanism, YoLoX-Nano, deep learning

## Abstract

Rails play a vital role in the bearing and guidance of high-speed trains, and the normal condition of rail components is the guarantee of the operation and maintenance safety. Fasteners are critical components for fixing the rails, so it is particularly important to detect whether they are in a normal state or not. The current rail-fastener detection models have some drawbacks, including poor generalization ability, large model volume and low detection efficiency. In view of this, an improved YoLoX-Nano rail-fastener-defect-detection method is proposed in this paper. The CA attention mechanism is added to the three output feature maps of CSPDarknet and the enhanced feature extraction part of the Path Aggregation Feature Pyramid Network (PAFPN); the Adaptively Spatial Feature Fusion (ASFF) is added after the PAFPN output feature map, which enables the semantic information of the high-level features and the fine-grained features of the bottom layer to be further enhanced. The improved YoLoX-Nano model has improved the AP value by 27.42% on fractured fasteners, 15.88% on displacement fasteners and 12.96% on normal fasteners. Moreover, the *mAP* value is improved by 18.75%, and it is 14.75% higher than the two-stage model Faster-RCNN on *mAP*. In addition, compared with YoLov7-tiny, the improved YoLoX-Nano model achieves 13.56% improvement on *mAP*. Although the improved model increases a certain amount of calculation, the detection speed of the improved model has been increased by 30.54 *fps* and by 32.33 *fps* when compared with that of the Single-Shot Multi-Box Detector (SSD) model and the You Only Look Once v3 (YoLov3) model, reaching 54.35 *fps*. The improved YoLoX-Nano model enables accurate and rapid identification of the defects of rail fasteners, which can meet the needs of real-time detection. Furthermore, it has advantages in lightweight deployment of terminals for rail-fastener detection, thus providing some reference for image recognition and detection in other fields.

## 1. Introduction

The operation and maintenance safety of high-speed railways is directly related to the national economic development. The rails play a key role in bearing and guiding high-speed trains, and the normal rail components are the guarantee of operation and maintenance safety. Fasteners are critical components for fixing the rail, so it is particularly important to detect whether the fastener is in a normal state [1]. With the rapid development of railway transportation in China, passenger and freight transport services have witnessed a dramatic increase, thus bringing more maintenance and management needs. Consequently, it is of great significance to ensure the safe operation of the railway infrastructure. As key components to fix the rails, fasteners are easily subject to displacement and fracture due to the large number and complicated stress imposed [2], which greatly affects the safety of the railway system. In this case, the detection of rail fasteners has become an important task to ensure safety. The detection of the health status of rail fasteners is generally carried out by professionals on a regular basis. However, the manual inspection method has the shortcomings of a long cycle and low efficiency. Moreover, as the number of railway lines and mileage covered grows, it fails to meet the high efficiency and accuracy requirements of the current railway inspection [3].

In order to solve the current problem, extensive studies have been carried out on the defect detection of rail fasteners at home and abroad, and thereby the method of image processing and recognition has been basically established as the main direction of the rail fastener detection. For instance, Liu J et al. [4] published a paper entitled “Learning Visual Similarity for Inspecting Defective Railway Fasteners”; Cui H et al. [5] published “Real-Time Geometric Parameter Measurement of High-Speed Railway Fastener Based on Point Cloud from Structured Light Sensors”; Santur Y et al. [6] put forward “Random Forest Based Diagnosis Approach for Rail Fault Inspection in Railways”; Fan H et al. [7] put forward “High-speed Railway Fastener Detection Using Minima Significant Region and Local Binary Patterns”; and Li Z W et al. [8] came up with “Identification of Temperature-Induced Deformation for HSR Slab Track Using Track Geometry Measurement Data”. Despite these models having the advantage of high computational efficiency, their generalization ability is poor, so targeted modifications are needed for different types of rail fasteners.

With the development of deep-learning image processing and target-detection technology, the deep-learning model is gradually applied to rail fastener-detection. For example, Qinan L et al. [9] proposed a model in their paper “Fine-grained Classification of Rail Fastener Images Based on B-CNN”, and Bai T et al. [10] put forward “An Optimized Railway Fastener Detection Method Based on Modified Faster R-CNN”. The iteration number is set to 4000 times in the training process in this paper. In the dataset in this paper, the precision of Faster R-CNN is 89.9%. Xiao L et al. [11] came up with “Missing Small Fastener Detection Using Deep Learning”; Chandran P et al. [12] proposed “An Investigation of Railway Fastener Detection Using Image Processing and Augmented Deep Learning”; and Zheng D et al. [13] proposed “A Defect Detection Method for Rail Surface and Fasteners Based on Deep Convolutional Neural Network”. All of these mentioned methods belong to the Two-Stage models. Such algorithms share the following common features. First, they obtain the target sub-regions where the image may be located, and then they extract target features from these sub-regions through the deep convolutional neural networks; finally, they perform the category detection and border correction. As a consequence, the Two-Stage models are mainly characterized by high accuracy and strong adaptability, while they have the disadvantages of large model volume and low detection efficiency [14,15]. On the other hand, Qi et al. [16] put forward “MYoLov3-Tiny: A New Convolutional Neural Network Architecture for Real-time Detection of Track Fasteners”. In the dataset of the article, the authors used 400 epochs, with the learning rate of 0.0001 and the last learning rate of 0.00001. The momentum is 0.9, and the weight decay is 0.0005 in two stages. In this dataset, the precision of Faster R-CNN is 78.5%, and the *mAP* is 90.5%. The precision of YoLov3 is 78.5%, and the *mAP* is 90.5%. MYoLov3-Tiny has a precision of 91.1%, *mAP* has a precision of 90.5% and YoLov3 has a precision of 99.3%.

Guo et al. [17] came up with “Real-time Railroad Track Components Inspection Based on the Improved YoLov4 Framework”. In the dataset of this paper, the authors used 200 epochs, with a learning rate of 0.00261, a batch size of 8 and a decay of 0.0005. For the dataset of this paper, the precision of YOLOv4 is 82.6%, and the precision of YOLOv4 improved by the authors is 84.2%. Wang et al. [18] presented “Real-Time Detection of Railway Track Component via One-Stage Deep Learning Networks”. In the dataset, YoLov3 has a *mAP* of 89% and a precision of 94%. Wan et al. [19] proposed railway tracks defects detection based on deep convolution neural networks, and Xiao et al. [20] proposed a track-fastener detection based on an improved YoLov4-Tiny Network. In the dataset of this paper, the *mAP* of the improved object detection network is 97.2%. These abovementioned algorithms belong to the One-Stage models, and such algorithms do not need to extract candidate regions and can directly detect the target categories and borders of images by using algorithms. Therefore, the main features of the One-Stage model are fast detection and small model volume but lower accuracy compared with the Two-Stage model. The One-Stage model has the advantages of fast detection speed and small size, which is more convenient for the subsequent deployment and application of the model, so it is favored in the engineering field [21]. For instance, Ma et al. [22] put forward “YoloX-Mobile: A Target Detection Algorithm More Suitable for Mobile Devices”. A YoloX-Mobile model was proposed to apply to the damage detection of conveyor belts based on its characteristics of small amount of calculation, strong reliability and generalization ability. Compared with the existing methods, the approach mentioned in that paper shows better reliability and convergence performance. Kou X et al. [23] put forward “Development of a YoLov3-Based Model for Detecting Defects on Steel Strip Surface”. The improved YoLov3 model is applied to the detection task of steel-strip surface defects. Moreover, the experiment shows that the improved model has higher detection accuracy than other models in GC10-DET dataset. In this case, One-Stage models have been extensively studied in various fields.

To sum up, the current Two-Stage models have high accuracy but low detection speed, which is not conducive to the application and model deployment in the railway industry with high real-time requirements. In contrast, the One-Stage models have high detection speed and small volume, which is convenient for deployment and real-time detection, but they have low accuracy. In order to achieve high precision and high efficiency in defective railway-fastener inspection, the paper proposes a new method based on the improved YoLoX-Nano model [24]. On the basis of the YoLoX-Nano model, three output feature maps of CSPDarknet and Path Aggregation Feature Pyramid Network (PAFPN) are used to strengthen the feature extraction part to increase the CA attention mechanism [25], which improves the detection performance of the model in the defective fasteners. Moreover, Adaptively Spatial Feature Fusion (ASFF) is added to the extracted part of the PAFPN network output, which learns the spatial filtering conflict information to suppress the inconsistency of the target image, so as to enhance the scale invariance of image features of the track fasteners and also reduce the computational cost required by the model during the reasoning process [26].

Based on the improved YoLoX-Nano model, the paper presents a novel method for the detection of rail fastener defects. The dataset of different defects of self-built rail fasteners under actual working conditions are preprocessed by data enhancement and other methods. According to the characteristics of the YoLoX-Nano model and the role of the coordinate attention (CA) mechanism, the CA attention mechanism is added to the three output feature layers of CSPDarknet and the feature network extraction part of PAFPN to improve the classification accuracy of the network model. In addition, in order to make full use of the semantic information of high-level features and the fine-grained features of the bottom layer, the ASFF is added after the output feature layer of PAFPN. On the premise of meeting the real-time detection of rail fastener defects, the detection accuracy and positioning accuracy of rail-fastener defects are improved. In addition, we also compared our model with the common YoLov4-tiny, YoLov3, YoLov5, Faster-RCNN, YoLov7-tiny and SSD network models trained on our dataset and used *mAP*, Precision, Recall, Size, *mMR*^−2^ and *Frame rate* as evaluation metrics for a comprehensive comparison. The improved model performs well in terms of detection precision, as well as localization precision, which can meet the practical needs of real-time defect detection of rail fasteners.

## 2. Materials and Methods

### 2.1. Data Acquisition and Preprocessing

The rail-fastener data used in this paper were mainly collected from the State Key Laboratory of Performance Monitoring and Protecting of Rail Transit Infrastructure. Mobile phones were used to shoot the photos of rail fasteners in the vertical direction of 10 to 100 cm from the rail fasteners on daily natural conditions, such as sunny, cloudy and rainy days, and a total of 746 pictures were obtained. Then 415 pictures were acquired through manual screening. According to the actual situation, the rail fasteners were divided into three categories, namely normal fasteners, displacement fasteners and fracture fasteners, including 144 normal ones, 135 displacement ones and 136 fracture ones, as shown in Figure 1. The filtered 415 images were labeled with Labelimg (an image labeling software) [27], and the labeled data were stored in the XML file in PASCAL VOC format.

Data enhancement is a commonly used method in the field of deep learning, with the purpose of improving the accuracy and generalization performance of the model and effectively improving the ability of the model to prevent overfitting [28]. In this experiment, a total of 415 image data were enhanced by image data enhancement, and the image data were amplified by a series of data enhancements such as flip, mirror, translation and scaling, and then they were randomly combined, as shown in Figure 2.

After data enhancement, 1290 images were finally obtained through manual screening, including 408 images of normal fasteners, 438 images of displacement fasteners and 444 images of fracture fasteners. In order to ensure the independence of the dataset, the corresponding fastener data of different forms were divided according to the proportion of training set, verification set and test set of 8:1:1. The sample distribution is shown in Table 1.

### 2.2. Principle of YoLoX-Nano Model

In the engineering application research of the target-detection algorithm, YoLo Series are favored by engineering researchers due to its characteristics of a fast response, high accuracy, simple structures and easy deployment [28]. The YoLo series converts the whole target detection task into the regression task for processing. The entire image is used as the input of the network, and then the location and category of the predicted box are predicted in the output layer.

In the previous YoLov3, YoLov4 and YoLov5, the Anchor-Based Approach is usually used. It means that the original input image will be meshed, and each feature point will correspond to some prior bounding boxes. The targets in each image are limited, and each anchor will produce many anchor boxes, thus producing a large number of easy samples; that is, the background box does not contain the target at all, and this can cause a serious imbalance between positive and negative samples. YoLoX, the latest improved version of the YoLo Series network models (as shown in Figure 3), has improved the convergence speed and accuracy of the model to varying degrees. The main improvements of YoLoX include the following:

Backbone: A focus module (as shown in Figure 4) is used to classify the feature point information and stack them on the channel.

In this module, a value is taken at each pixel interval in an image, and four independent feature layers are obtained. The four maps are then stacked so that the width information and height information are concentrated to the channel information, thus making the input channels expand four times and reducing the number of parameters to be calculated.

The YoLoX model uses the SPP network structure in the CSPDarknet backbone extraction network. The SPP network structure mainly pools the feature maps by pooling kernels of different sizes and then merges the pooled results to enhance the perceptual wildness of the CSPDarknet backbone network. The SPP network structure is shown in Figure 5.

The partial mathematical definition of the SPP network structure is shown in Equations (1) and (2):(1)$$\mathrm{Kernel}\;\mathrm{size}:\;\lceil \frac{h_{in}}{n},\frac{w_{in}}{n}\rceil =ceil\left(\frac{h_{in}}{n},\frac{w_{in}}{n}\right)$$
(2)$$\mathrm{Stride}\;\mathrm{size}:\;\lceil \frac{h_{in}}{n},\frac{w_{in}}{n}\rceil =floor\left(\frac{h_{in}}{n},\frac{w_{in}}{n}\right)$$

In Equations (1) and (2), *c*, *h_in_* and *w_in_* represent the number of channels, height and width of the input image, respectively; (*n*, *n*) represents the number of pooling; the *ceil()* function represents the number rounded up to the nearest number; and the *floor()* function represents the number rounded down to the nearest number.

To achieve a better feature extraction effect in the YoLoX model, the nonlinear activation function SiLU is used in the CSPDarknet backbone extraction network. The SiLU activation function is based on the optimization of the Sigmoid function and the ReLU activation function, which has a stronger nonlinear capability than the ReLU activation function and solves the problem that the output in the ReLU activation function is 0 when there is a negative input, resulting in the problem of gradient dispersion. It also inherits the advantage of a faster convergence of the ReLU activation function. A comparison of the function images is shown in Figure 6.

The mathematical definition of the ReLU and SiLU activation functions is shown in Equations (3) and (4):(3)ReLU=max(0,x)
(4)$$\mathrm{SiLU}=xsigmiod\left(x\right)=x\left(\frac{1}{1+e^{-x}}\right)$$

In the classification layer and regression layer, Decoupled Head serves as the detection head. The Decoupled Head used in the previous version of YoLo was implemented by classification and regression in a 1 × 1 convolution kernel, which adversely affected the recognition of the network. In YoLoX, the Obj and Cls branches use the binary cross-entropy loss function (BCELoss), and the Reg branch uses IouLoss, so the YoLoHead is divided into two parts, discriminative classification and quantitative regression, to implement their respective functions, as shown in Figure 7.

The mathematical definition of YoLo Head is shown in Equations (5) and (6):(5)$$L_{cls}=-\overset{n}{\underset{i=1}{\sum }}\left(t_{i}\log(p_{i}\right)+\left(1-t_{i}\right)\log(1-p_{i}))$$
(6)$$L_{reg}=-\log(IOU\left(B_{gt},B_{pred}\right))$$
where *t_i_* represents the category to which the *i*-th sample belongs, *p_i_* represents the probability of belonging to the category, *L_cls_* represents classification loss, *L_reg_* represents localization loss, *B_gt_* is the ground truth bounding box and *B_pred_* is the predicted bounding box.

The classification part is used to acquire the category information of detection objects, while the regression part is used to obtain the detection box information and confidence information of the rail fasteners. Then the final prediction integrates this information together, which can be reverse optimized according to the loss function to improve the model accuracy.

In terms of data enhancement, Mosaic is used in YoLoX. The principle of Mosaic is to use four images for the Concat splicing so as to achieve data enhancement. Moreover, the data of four images are calculated simultaneously in the BN calculation to enrich the background information of the detected objects.

YoLoX adopts the principle of Anchor Free, whose biggest advantage is the fast detection speed. With the Anchor Free, there is no need to preset anchors; it only needs to regress the target center point and width and height of the feature map of different scales, thus greatly reducing the time consumption and computational effort. In addition, YoLoX also uses simOTA in dynamic matching of the number of positive samples, which can automatically determine the feature map to be matched. The calculation process is shown in Figure 8.

The simOTA method of dynamically matching positive samples avoids the problems of inaccurate localization of multiple targets and excessive generalization ability of detection boxes due to Anchor Free. Because the images of rail fasteners are taken from different heights, and the size of the fasteners varies, the detection box of the model needs to adapt to multiscale detection. Therefore, the dynamic matching method of simOTA is more suitable for the detection of rail fasteners in this dataset.

### 2.3. The Improved YoLoX-Nano Model

The improvement of the YoLoX-Nano model is mainly reflected in two aspects, as shown in Figure 9.

After the three feature layers of the backbone are extracted, the CA attention mechanism is introduced with PAFPN to strengthen feature extraction, which will make the network focus more on the target to be detected and ignore the unnecessary background information.

In order to make full use of the semantic information of high-level features and the fine-grained features of the bottom layer, ASFF is added to the output part of the feature-extraction network strengthened by PAFPN. The ASFF can adaptively learn the spatial weight of feature-mapping fusion at different scales, and the features of different layers are fused together by learning the weight parameters, which can reduce the interference of invalid features to the target detection.

#### 2.3.1. Principle of Adding CA Attention Mechanism

Similar to the human perceptual process, the coordinate attention mechanism can make the model selectively focus on a certain part of information, so as to alleviate the information overload and enable the model to process more important information more efficiently [25]. The core idea of the coordinate attention mechanism is to make the model selectively obtain the main differences of each rail-fastener feature image and put the main calculation process into the tasks related to rail-fastener-defect detection. Then the results obtained from the tasks are used to reversely regress to the weight center of the feature map in order to efficiently complete the rail fastener detection task. 

In this paper, the CA attention mechanism is different from the channel attention mechanism in which the feature tensor is converted into a single feature vector. The CA attention mechanism decomposes the channel attention into two one-dimensional feature-encoding processes, which perform pooling and convolution operations along the width and height dimensions of the feature layer, so as to obtain the feature encoding. Moreover, the encoding processes aggregate features along the two spatial directions, respectively. In this way, remote dependencies can be captured along one spatial direction, while precise positional information can be preserved along the other spatial direction. The resulting feature maps are then encoded separately into a pair of direction-aware and position-sensitive attention maps that can be complementarily applied to the input feature maps to enhance the representations of the objects of interest [25]. 

Specifically, the CA attention mechanism performs average pooling in both horizontal and vertical directions. Through spatial information coding, spatial information is integrated in a weighted manner, so that the model can accurately locate the position and identify the category of rail fasteners and identify the target area.

The CA attention mechanism module is shown in Figure 10. For the part where the image is embedded in the coordinate information, the CA attention mechanism performs the global average pooling for the given input image data in the horizontal and vertical directions. Moreover, the global average pooling is converted into the operation of one-dimensional feature encoding, making it easier for attention module to accurately capture the location information. For input tensor x, using pooling kernel sizes of (*h*, 1) and (1, *w*) to encode horizontal and vertical features, the calculation formulas are shown in Equations (7) and (8):(7)$$Z_{c}^{h}\left(h\right)=\frac{1}{w}\underset{0\leq i<w}{\sum }x_{c}\left(h,i\right)$$
(8)$$Z_{c}^{w}\left(w\right)=\frac{1}{h}\underset{0\leq j\leq w}{\sum }x_{c}\left(j,w\right)$$
where Equation (7) represents the output of the *c*-th channel at height, *h*; and Equation (8) represents the output of the *c*-th channel at width, *w*.

The average pooling in these two directions is then spliced by Concat, and the channels are compressed by using the convolution transform so that the number of channels ranges from C to C/r. Moreover, r is used to control the compression reduction rate. The batch normalization and the nonlinear activation function are used to process the spatial information encoded in the both the vertical direction and the horizontal direction. The calculation formula is shown in Equation (9):(9)$$f=\partial \left(F1(\left[z^{h},z^{w}\right])\right)$$

In Equation (3), *F*1 denotes the one-dimensional convolution kernel; *z^h^*, *z^w^* represents the Concat splicing operation; and ∂ denotes the nonlinear activation function.

The main purpose of $F_{h}$ and $F_{w}$ here is to convert $f_{h}$ and $f_{w}$ into tensors with the same number of channels as the input x, using two 1×1 convolution kernels. After the Sigmoid activation function, the attention weight, *g^h^*, of the feature map in the height and width and the attention weight, *g^w^*, in the width direction are obtained, respectively, as shown in Equations (10) and (11):(10)$$g^{h}=\sigma \left(F_{h}\left(f^{h}\right)\right)$$
(11)$$g^{w}=\sigma \left(F_{w}\left(f^{w}\right)\right)$$

After the above calculation, the attention weigh, *g^h^*, in the height direction and the attention weight, *g^w^*, in the width direction of the input feature map are obtained. The final feature map with attention weights in the width and height directions is then obtained by multiplicative weighting calculation on the original feature map. Finally, the output *y_c_* (*i*,*j*) of the CA module is calculated as shown in Equation (12):(12)$$y_{c}\left(i,j\right)=x_{c}\left(i,j\right)\times g_{b}^{h}\left(i\right)\times g_{c}^{w}\left(j\right)$$

In order to expand the receptive field, the SPP model is added to the deep network in the YoLoX-Nano model, and the PAFPN model is used to achieve deep fusion of features, so as to obtain remarkable features of rail fasteners in different output layers. Since the channels in deep networks mainly retain abstract semantic information, a single channel becomes an independent individual to preserve information. Therefore, this paper studies the weights of the reassigned channel after employing the CA attention mechanism in the PAFPN modules. Moreover, the addition of the CA attention mechanism to the three feature layers of the backbone feature-extraction network is also studied. The location of the CA attention mechanism is added between the modules, as shown in Figure 11.

#### 2.3.2. Principle of Adding ASFF Module

In order to fully take advantage of the semantic information of high-level features and the fine-grained features of the bottom layer, most networks will output multiple feature maps through FPN [26]. In the YoLoX-Nano, PAFPN is used to fuse the deep feature information of rail fasteners. This method of fusion only simply transforms the output feature maps into the same size by convolution and then combines them sequentially, which cannot fully utilize the features at different scales. Consequently, the ASFF is introduced between the PAFPN and the YoLoHead. The core principle of ASFF is to adaptively learn the spatial weight of fusion for feature maps at each scale, and features of different layers are fused together by learning the weight parameters. In this way, the inconsistency in feature pyramids can be addressed, so as to retain the useful information for combination. The addition position of ASFF is shown in Figure 12.

In Figure 12, Out1, Out2 and Out3 are three feature layers output from the PAFPN. The blue box represents how the features are fused in the ASFF. $X^{1\to 3}$, $X^{2\to 3}$ and $X^{3\to 3}$ denote the features of Out1, Out2 and Out3 feature layers, respectively. Then we multiply $X^{1\to 3}$, $X^{2\to 3}$ and $X^{3\to 3}$ by their respective weights and sum them up. Finally, the feature ASFF-3 is acquired after the feature fusion. The principle of the specific operation is shown in Equation (13):(13)$$y_{ij}^{l}=\alpha^{3}\cdot X_{ij}^{1\to l}+\beta^{3}\cdot X_{ij}^{2\to l}+\gamma^{3}\cdot X_{ij}^{3\to l}$$
where $y_{ij}^{l}$ denotes the new feature map, while *α*^3^, *β*^3^ and *γ*^3^ denote the weight parameters of the three feature maps respectively. In addition, $X_{ij}^{1\to l}$, $X_{ij}^{2\to l}$ and $X_{ij}^{3\to l}$ denote features of the Layer 1, Layer 2 and Layer 3, respectively, making it satisfy $\alpha^{3}+\beta^{3}+\gamma^{3}=1$ by the Softmax function and define Equation (14). Here $\lambda_{\alpha }^{3}$, $\lambda_{\beta }^{3}$ and $\lambda_{\gamma }^{3}$ serve as control parameters:(14)$$\alpha^{3}=\frac{e^{\lambda_{\alpha }^{3}}}{e^{\lambda_{\alpha }^{3}}+e^{\lambda_{\beta }^{3}}+e^{\lambda_{\gamma }^{3}}}$$

### 2.4. Methods for Model Evaluation

#### 2.4.1. Environment Setting for Modeling

In this study, the experimental platform is Windows10 operating system, the deep learning framework is Pytorch1.8.1 and the CPU is 11th Gen Intel i5-11400 H with RAM of 16 GB. The Graphics card is NVIDIA GeForce GTX1650 Laptop 4G, and CUDA11.1 is used as the parallel computing framework at the bottom layer.

In terms of the training strategies, the image size is 640 × 640, the Batch Size is set to 8 and the learning rate is adjusted by using cosine annealing decay. A total of 100 iterations of the training dataset are trained without freezing training. The initial learning rate is 0.001, and the learning decay rate is 0.92. In order to verify the feasibility of the improved model on rail fasteners and its effectiveness on the rail-fastener detection, this paper compares several schemes in the research and design: (1) the effects of the improvement schemes of different modules of YoLoX-Nano on the mean average precision, the precision and the recall rate of the rail fastener detection; and (2) the current improved model has promoted in mean average precision, average precision, recall rate and detection speed compared with the common One-Stage YoLov3, SSD, YoLov5 models and two-stage algorithm Faster-RCNN model. 

#### 2.4.2. Model Evaluating Indicators

The paper mainly uses the detection accuracy of the model and the detection speed of rail fasteners to evaluate the performance of the model in the test set. Testing indicators include *P* (precision), *R* (recall), *AP* (average precision), *mAP* (mean average precision), *F*1 Score and *MR*^−2^ (log Average miss rate). The *AP* is a comprehensive consideration of precision and recall, which can be used to measure the reliability of the detection algorithm and is also an intuitive evaluation criterion for determining the accuracy of the detection model. The *mAP* reflects the comprehensive performance of the detection model in identifying all categories. The *F*1 Score is a weighted average sum of the precision and recall of the detection model, and it is also an index to measure the accuracy of the detection model [29,30]. Considering that the paper focuses primarily on multi-category detection, the three categories of rail fasteners should be averaged by using *MR*^−2^, so as to comprehensively judge the performance of the model, and *mMR*^−2^ is obtained. Moreover, *fps* (frames per second) is also used as an evaluation indicator of the model detection speed. The evaluation indicators are calculated as follows:(15)$$P=\frac{TP}{TP+FP}\times 100\%$$
(16)$$R=\frac{TP}{TP+FN}\times 100\%$$
(17)$$AP=\int_{0}^{1}P\left(R\right)dR$$
(18)$$mAP=\frac{1}{n}\overset{n}{\underset{i=1}{\sum }}AP_{i}$$
(19)$$FPS=\frac{n}{t}$$
(20)$$F1=\frac{2\times P\times R}{P+R}$$

In the Equations, *P* denotes precision, *R* denotes recall, *TP* denotes the number of true prediction and *FP* denotes the number of positive samples of false prediction. Furthermore, *FN* denotes the number of negative samples of false prediction, *n* denotes the number of target categories to be detected, *AP_i_* denotes the average precision of the *i*th target category, *N* denotes the number of images to be detected, *t* denotes the detection time and *F*1 denotes the harmonic mean of precision and recall [31].

Wherein the evaluation indicator *MR*^−2^ is used to quantify the *MR-FPPI* curve, *MR* and *FPPI* are mutually exclusive indicators. When the threshold value of the model is low, the model detects more targets, with less missed detection but higher false detection. On the contrary, the false detection samples decrease and the missed detection samples increase when the threshold value increases. The *MR-FPPI* curve is acquired through setting different threshold values. The calculation of *MR*^−2^ is more complicated, with the *FPPI* value as the horizontal coordinate and the *log(MR)* value as the vertical coordinate. A total of 9 values of *FPPI* are randomly obtained in the range of [0.01, 1], and the corresponding vertical coordinate values are obtained and averaged. Lower values of *MR*^−2^ represent better performance of the model, and the calculation formulas are as follows:(21)MR=1−Re c all
(22)$$FPPI=\frac{FP}{N}$$
(23)$$MR^{-2}=\frac{1}{9}\sum \delta \left(FPPI\right)$$
(24)$$mMR^{-2}=\frac{1}{C}\overset{C}{\underset{0}{\sum }}MR^{-2}$$
where *MR* denotes missed detection rate, *FPPI* denotes average false detection rate of the images, *N* denotes the number of pictures, δ(*FPPI*) denotes the mapping relationship between *FPPI* and *log*(*MR*)**, and *C* denotes the number of categories.

## 3. Results

### 3.1. Comparison of Different Module Combination Schemes

In the paper, YoLoX-Nano serves as the basic comparison network on which the CA attention mechanism is added to the three output feature layers of CSPDarknet and the enhanced feature extraction part of PAFPN. Moreover, the ASFF module is added between the PAFPN output feature layer and YoLoHead for improvement, and some groups of comparison models are trained, as shown in Table 2. Among them, the CA attention mechanism is added to the three feature output layers in the CSPDarknet, and the ASFF module is added to PAFPN output feature layer, which is taken as the final improvement scheme of this paper. Under the same training conditions, the results of model identification are shown in Table 2. Compared to the original YoLoX-Nano, the improved model proposed in the paper has an increase of 21.05% in *mAP* and a decrease of 0.2887 in *mMR*^−2^. The identification accuracy of rail fasteners is mainly expressed by recall and precision.

After the training is completed, the Loss value is plotted through the curve, in which the training process can be directly observed. Figure 13 depicts the loss curves of the original YoLoX-Nano network and its improved version during the training process. It can be seen from Figure 13 that, after the network is trained for about 95 times, the parameters tend to be stable, and the loss value of the original YoLoX-Nano and the improved one decreases to about 1.7 and 1.09, respectively. The analysis of the convergence of loss function shows that the improved YoLoX-Nano network outperforms the original network in terms of converging speed, indicating a relatively ideal training result.

By comparing the original YoLoX-Nano model and the improved model in Table 2, it can be seen that the *mAP* value difference between the unimproved YoLoX-Nano model and the improved model is about 18.75% in the detection accuracy. From Figure 13, it is clear that a relatively huge gap exists in the actual detection effects between the original model and the improved model. Furthermore, some errors occur when the unimproved model detects the defective components. By comparison, the improved model has higher prediction scores and detection accuracy in displacement and crack detection. Figure 14 depicts the practical detection results of the unimproved model and the improved model.

According to the comparison of the detection results Figure 14, the improved YoLoX-Nano model can accurately detect the rail-fastener offset defects with high confidence. In contrast, the original model has low confidence in the detection of fastener offset and has errors in classification. Moreover, when it comes to the detection of normal and crack fasteners, the improved model has higher accuracy than that of original one. Consequently, the improved YoLoX-Nano performs better than its original one in terms of detection accuracy, classification accuracy and localization accuracy.

### 3.2. Comparison of Comprehensive Performance of Various Network Models

In the track-fastener detection direction, Bai T et al. [10] put forward the Modified Faster R-CNN. The iteration number is set to 4000 times in the training process in this paper. The dataset in this paper, the precision of the Faster R-CNN, is 89.9%. Qi H et al. [16] put forward MYoLov3-Tiny. In this dataset, the precision of the Faster R-CNN is 78.5%, and the *mAP* is 90.5%. The precision of YoLov3 is 78.5%, and the *mAP* is 90.5%. MYoLov3-Tiny has a precision of 91.1%, *mAP* has a precision of 90.5%, and YoLoV3 has a precision of 99.3%. Wang T et al. [18] put forward the Improved YoLov3. In the dataset, YoLoV3 has a *mAP* of 89% and a precision of 94%. Xiao T et al. [20] put forward the YoLov4-Tiny Network. In the dataset of this paper, the *mAP* of the improved object detection network is 97.2%. So, in order to verify the comprehensive performance of the improved YoLoX-Nano network model, a comparison with the SSD model, Faster-RCNN, YoLov3, YoLov5, YoLov4-tiny and YoLov7-tiny model was established in this paper.

According to Table 3, the improved YoLoX-Nano model increased the *mAP* value by 2.71% and the detection speed by 30.54 *fps* to 54.35 *fps* over the SSD model under the same conditions for the dataset of this study, compared to YoLov4-tiny, YoLov3, YoLov5 and YoLov7-tiny, which increased the *mAP* by 37.21%, 33.72%, 15.73% and 1.51%, respectively, and compared to YoLov3 and YoLov5 increased the detection speed by 32.33 *fps* and 7.08 *fps*. The detection speed is improved by52.35 *fps*, and the *mAP* is improved by 14.75% compared with the two-stage algorithm Faster-RCNN, and the *MR*^−2^ value is reduced by 0.2434 compared with the Faster-RCNN model, which also has a significant advantage over other algorithms in terms of the number of model parameters. Although the detection speed is 28.89 *fps* and 2.11 *fps* lower than that of YoLov4-tiny and YoLov7-tiny, respectively, the detection speed of 54.35 *fps* meets the requirements of rail fasteners detection. In conclusion, the improved YoLoX-Nano model has better comprehensive performance.

## 4. Conclusions

In order to improve the detection efficiency and accuracy of rail fastener defects, an improved YoLoX-Nano method based on convolutional neural network is proposed in this paper. The CA attention mechanism is added to the backbone feature-extraction network output and to the PAFPN deep feature fusion, which effectively addresses the problem of information overload, as well as the prioritization of more important information. Moreover, the ASFF is added to the output end of PAFPN, so that the model can adaptively learn the spatial weight of fusion for feature maps at various scales, and features of different layers are fused together by learning the weight parameters, thus solving the problem of inconsistency in the feature pyramid. The improved YoLoX-Nano model proposed in the paper has higher precision, recall and confidence in rail-fastener defect detection compared with the YoLoX-Nano model before improvement and the SSD, YoLov3, YoLov5 and Faster-RCNN models. The improved model performs well in terms of detection precision, as well as localization precision, which can meet the practical needs of real-time defect detection of rail fasteners. This research provided a high efficiency and accuracy detection method to ensure railway track safety and safe operation and maintenance of the high-speed railway.

## Figures and Tables

**Figure 1 sensors-22-08399-f001:**
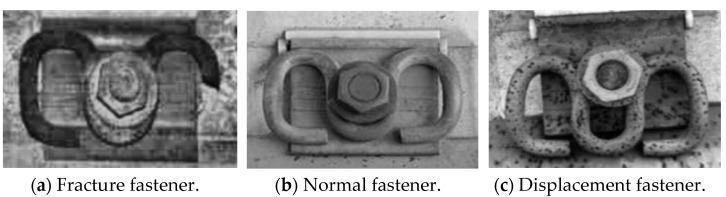
Forms of rail fasteners.

**Figure 2 sensors-22-08399-f002:**
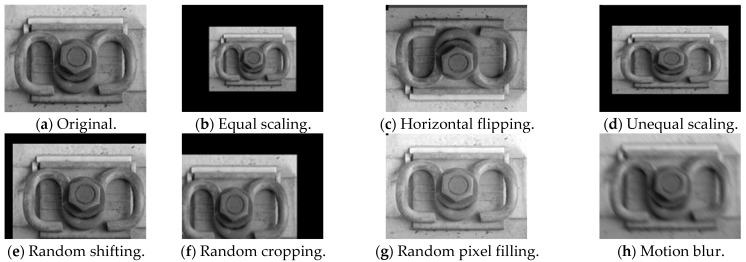
Image data enhancement with different enhanced methods.

**Figure 3 sensors-22-08399-f003:**
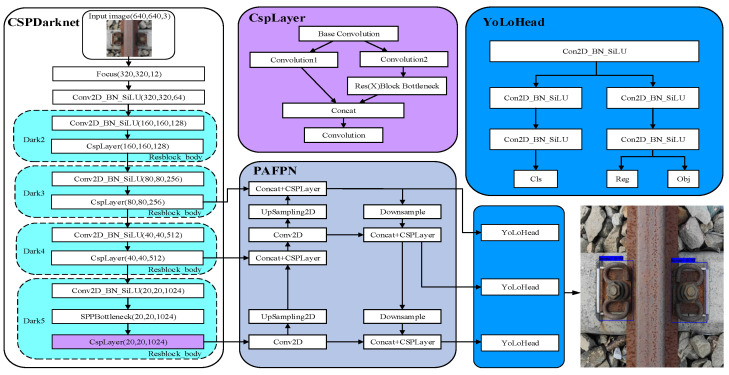
YoLoX-Nano neural network model.

**Figure 4 sensors-22-08399-f004:**
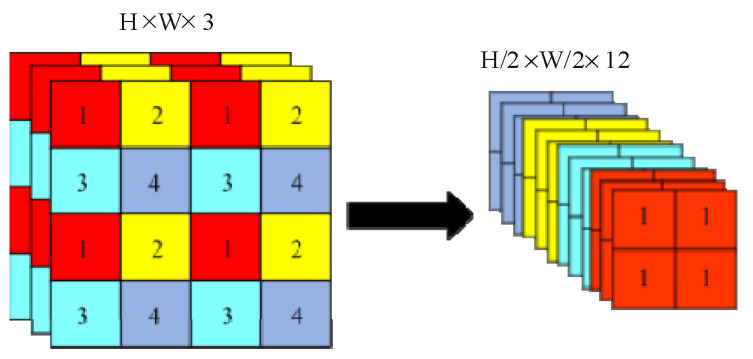
The focus module.

**Figure 5 sensors-22-08399-f005:**
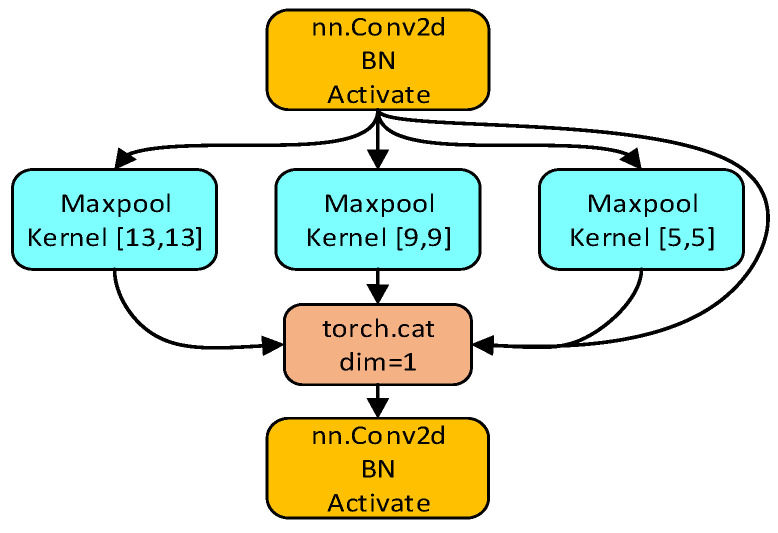
SPP network structure.

**Figure 6 sensors-22-08399-f006:**
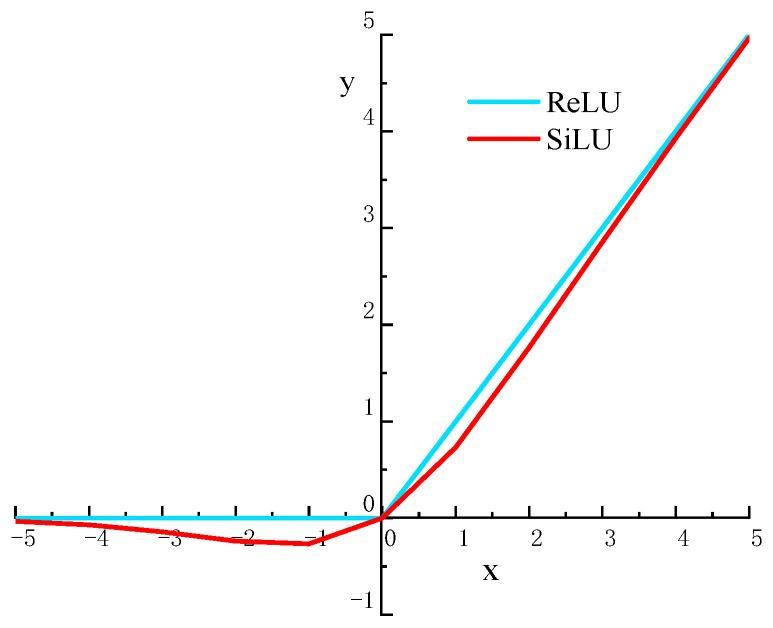
Function image comparison.

**Figure 7 sensors-22-08399-f007:**
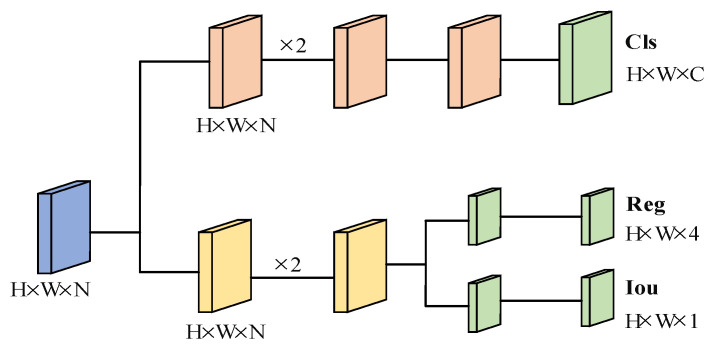
YoLoX-Nano Decoupled Head.

**Figure 8 sensors-22-08399-f008:**
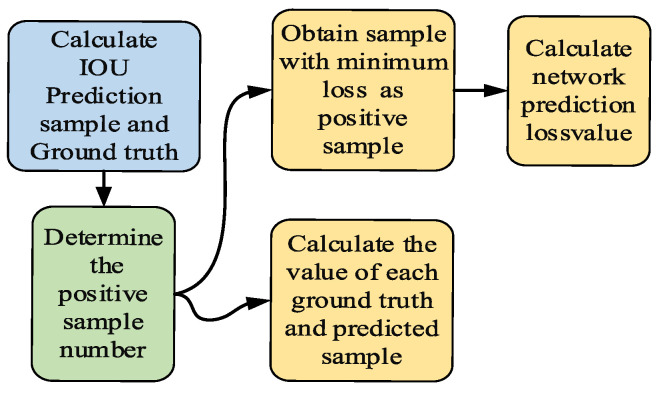
The computational processes of simOTA.

**Figure 9 sensors-22-08399-f009:**
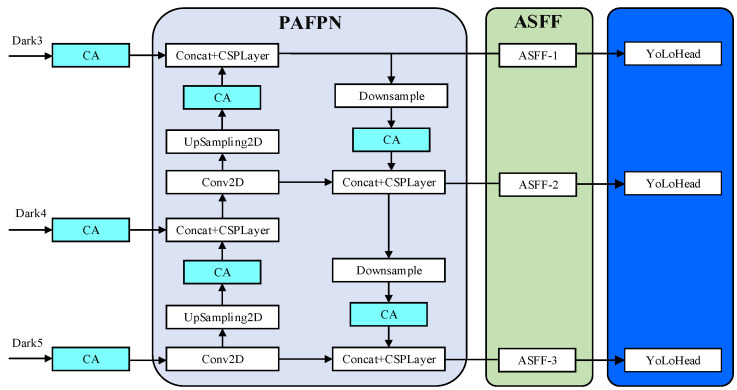
Improvement scheme of YoLoX-Nano network model.

**Figure 10 sensors-22-08399-f010:**
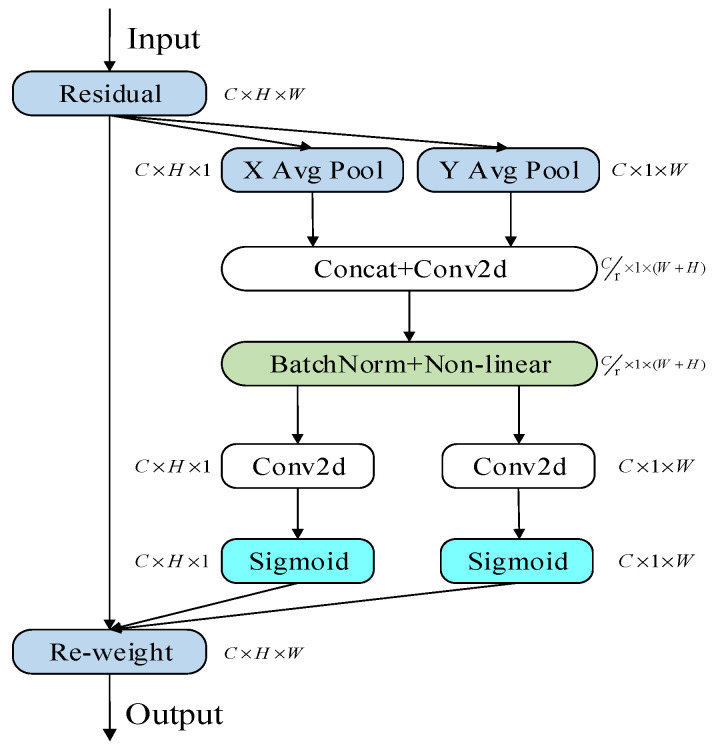
Coordinate attention (CA).

**Figure 11 sensors-22-08399-f011:**
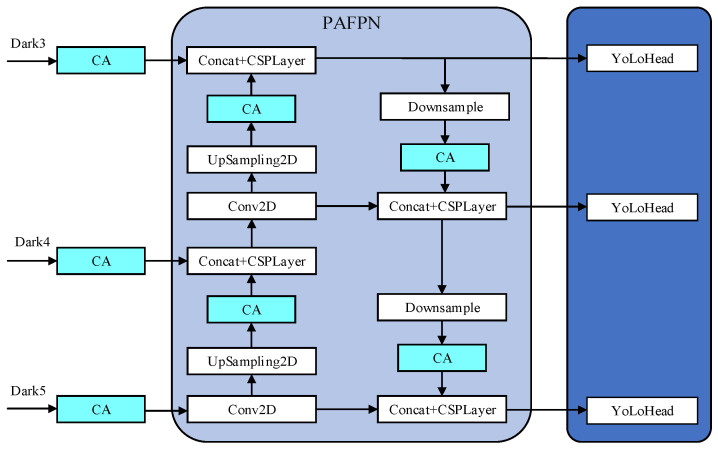
The PAFPN module after being improved.

**Figure 12 sensors-22-08399-f012:**
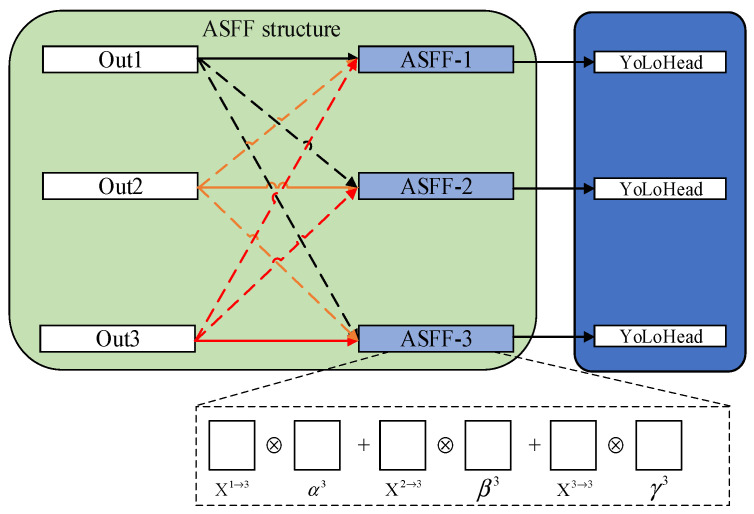
The ASFF model after being improved.

**Figure 13 sensors-22-08399-f013:**
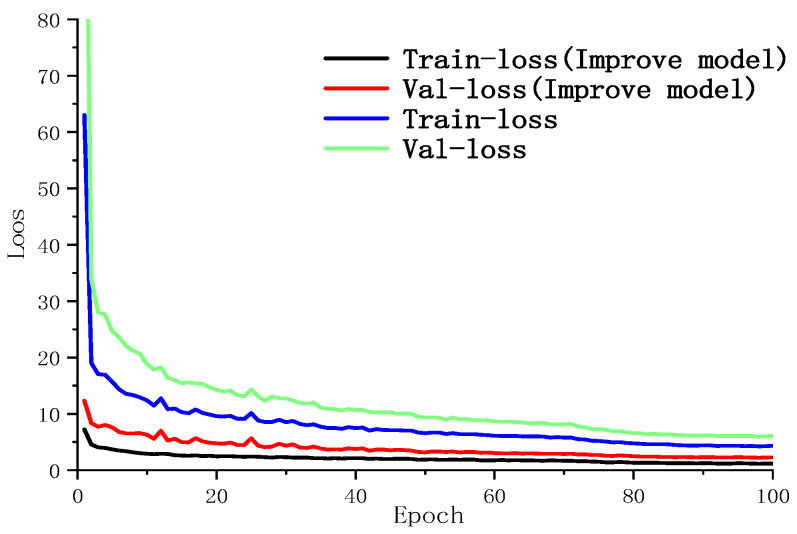
YoLoX-Nano loss function curve before and after improvement.

**Figure 14 sensors-22-08399-f014:**
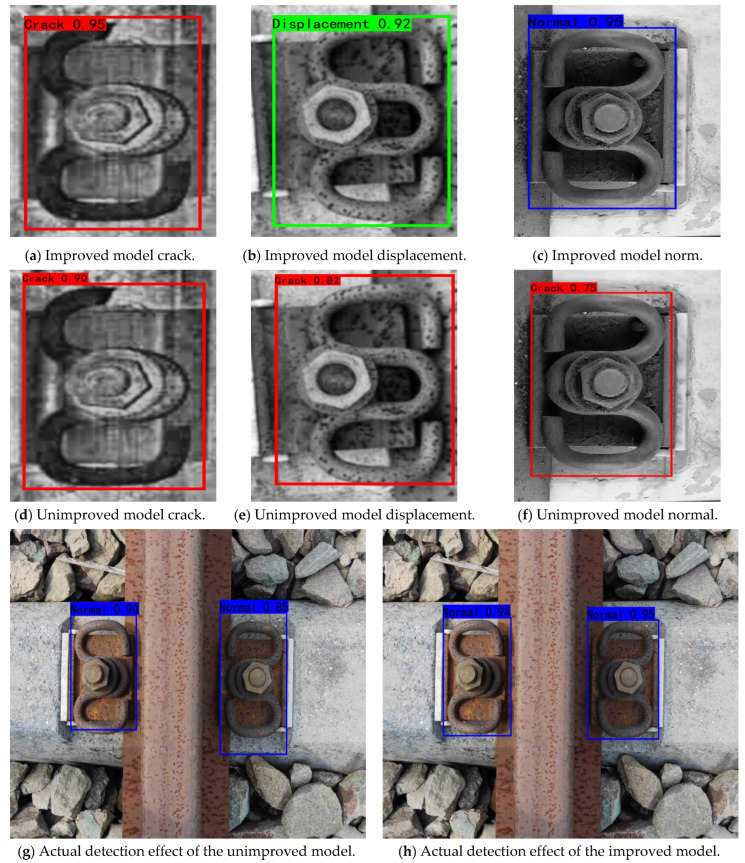
Actual detection images of the unimproved model and improved model.

**Table 1 sensors-22-08399-t001:** Data sample distribution.

Dataset	Normal	Displacement	Crack
Training set	326	350	355
Validation set	41	44	44
Test set	41	44	45

**Table 2 sensors-22-08399-t002:** Comparison of detection effects of various improvement schemes.

Scheme	Crack		Displacement		Normal		*mAP*/%	*mMR* ^−2^
	*AP/%*	*mMR* ^−2^	*AP/%*	*mMR* ^−2^	*AP/%*	*mMR* ^−2^		
1	66.95	0.45	83.96	0.24	87.04	0.14	79.32	0.277
2	93.04	0.11	95.77	0.06	95.79	0.05	94.86	0.073
3	88.84	0.16	100	0	100	0	96.28	0.0533
4	94.37	0.08	99.84	0	100	0	98.07	0.0266

1: YoLoX-Nano, 2: CA + PAFPN, 3: CA + ASFF, 4: CA + PAFPN + ASFF.

**Table 3 sensors-22-08399-t003:** Performance comparison of various network models.

Networks	*mAP*/%	Precision	Recall	Size (KB)	$mMR^{-2}$	Frame Rate (*fps*)
YoLov4-tiny	60.86	63.16	24.9	23032	0.51	**83.42**
YoLov3	64.35	63.9	74.7	240735	0.143	22.02
YoLov5	82.34	97.6	12.2	27822	0.28	47.27
Faster-RCNN	83.32	97.6	30.1	110854	0.27	2.3
YoLov7-tiny	84.51	96.7	23.4	23695	0.246	56.46
SSD	96.56	96.2	**95.5**	102702	0.063	23.81
Improved-YoLoX-Nano	**98.07**	**97.7**	93.4	**9289**	**0.0266**	54.35

The font is bold to reflect the advantages of each model.

## Data Availability

The data presented in this study are available upon request from the corresponding author.

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
