# Peer review of "High Speed Railway Fastener Defect Detection by Using Improved YoLoX-Nano Model"

_sensors, 2022, doi:10.3390/s22218399_

Round 1
Reviewer 1 Report
1. The contributions of modules CA, PAFPN, ASFF, etc. to deep learning models have all been verified in the paper [23][22][24], so the authors need to further clarify and verify what the innovative work of this paper contributes to the performance improvement of the model.
2. The authors need to further clarify the relationship and differences between the work in this paper and the literature [23][22][24], and the description of the reuse and validation of the published work should be reduced.
3. The authors have explained in the article that YOLOX is more suitable for railway defect detection than YOLOv3 and YOLOv5, so why should YOLOv3 and YOLOv5 be compared in comparison experiments? Moreover, comparative experiments with the various algorithms mentioned in the Related Work section of the paper should be added.
4. Since the resolution of Figure 14 is relatively low and the numbers in the figure cannot be seen clearly, so it is recommended to replace a high-quality original image. In addition, the image layout is too loose and needs to be adjusted.
5. Please check the correctness of the letter size of "YoLoX" in the whole manuscript.
Reviewer 2 Report
Major:
1. The title can be improved, “High Speed Railway Fastner Defect Detection Using Improved CNN Based YoloX-Nano Model”.
2. Line 98: The advantage of the One-Stage models in fast detection with supporting it with literature examples. However, the Two - Stage models need to be supported to with advantages and disadvantages to support the sacrifice of the accuracy for performance using One-Stage models. As stated in Line 86, but with more discussion.
3. The related literature or related work section is missing. You can move the previous literature discussion to the related work section. So, you keep the introduction concise and focus on describing the main problem you are solving.
4. The Literature discussion is limited only on the used algorithm. It will be better if you discuss the previous research in more details, such as, mentioning the used model’s configuration, hyper parameters, and results of both accuracy and performance. So, the significance of your work can be clearer to the reader.
5. Line 176: The mentioned data splitting is 8:1:1, did you try another ratio such as 70:15:15. As the data splitting can have an effect on the model’s training, especially high training data may lead to overfitting. For that, was there a test for the model’s accuracy with different splitting ratio?
6. When splitting the data, was it random samples for each class?
7. Did you consider using YoloR or Yolo V7 model and compare their accuracy and efficiency in your study?
YoloR and Yolo v7 are more accurate in general than YoloX. However, in your application, it is interesting to compare the computation time.
8. SiLU was used as the activation function. SiLU is the default activation function for the YoloX models, but it would be better to compare it to other leaky Relu variants in your application.
9. Line 399: need just a simple justification for using SoftMax function.
10. Line 410: why using exactly 100 epochs for training the model? Why did not you use early stopping to avoid overfitting of the model?
Minor:
1. The abstract is a bit wordy; it should be more concise so the reader can grasp the whole research article without much time.
2. Line 105: reference number (20). It is true that YoloX model was used in this research, but it does not feel that it is related or needed as a reference to your work.
3. Add article’s organization paragraph by the end of the introduction section.
4. Equations’ 7, 8, 10, and 11 need more clarification for the parameters.
5. Formatting of equation 14 to 19 and 20 to 23 need to be changed for clear reading.
6. Figure 14’s sub figures need to be reformatted as in the current format there are a lot of wasted space in the page.
7. It would be better to list of abbreviations for the readers.
Round 2
Reviewer 1 Report
Please check the references further to ensure that the most representative and up-to-date literature in the relevant field is perfectly cited. Moreover, the English needs further polishing.